# Antibiotic Utilization Patterns for Different Wound Types among Surgical Patients: Findings and Implications

**DOI:** 10.3390/antibiotics12040678

**Published:** 2023-03-30

**Authors:** Zikria Saleem, Umar Ahsan, Abdul Haseeb, Ummara Altaf, Narjis Batool, Hira Rani, Javeria Jaffer, Fatima Shahid, Mujahid Hussain, Afreenish Amir, Inaam Ur Rehman, Umar Saleh, Sana Shabbir, Muhammad Usman Qamar, Waleed Mohammad Altowayan, Fahad Raees, Aisha Azmat, Mohammad Tarique Imam, Phumzile P. Skosana, Brian Godman

**Affiliations:** 1Department of Pharmacy Practice, Faculty of Pharmacy, Bahuddin Zakaria University, Multan 60800, Pakistan; 2Department of Infection Prevention and Control, Alnoor Specialist Hospital, Ministry of Health, Makkah 24241, Saudi Arabia; 3Department of Clinical Pharmacy, College of Pharmacy, Umm Al-Qura University, Makkah 24382, Saudi Arabia; 4Department of Pharmacy, Ghurki Trust Teaching Hospital, Lahore 54000, Pakistan; 5Center of Health Systems and Safety Research, Faculty of Medicine, Health and Human Sciences, Australian Institute of Health Innovation, Macquarie University, Sydney 2109, Australia; 6Department of Pharmacy, Faculty of Pharmacy, University of Lahore, Lahore 54000, Pakistan; 7Department of Pharmacy, Indus Hospital and Health Network, Karachi 75190, Pakistan; 8Department of Microbiology, Armed Forces Institute of Pathology, National University of Medical Sciences, Rawalpindi 46000, Pakistan; 9Punjab University College of Pharmacy, Faculty of Pharmacy, University of the Punjab, Lahore 54000, Pakistan; 10Institute of Microbiology, Faculty of Life Sciences, Government College University Faisalabad, Faisalabad 38000, Pakistan; 11Department of Pharmacy Practice, College of Pharmacy, Qassim University, Buraydah 52571, Saudi Arabia; 12Department of Medical Microbiology, Faculty of Medicine, Umm Al-Qura University, Makkah 24382, Saudi Arabia; 13Department of Physiology, Faculty of Medicine, Umm Al-Qura University, Makkah 24382, Saudi Arabia; 14Department of Clinical Pharmacy, College of Pharmacy, Prince Sattam Bin Abdul Aziz University, Al Kharj 11942, Saudi Arabia; 15Department of Clinical Pharmacy, School of Pharmacy, Sefako Makgatho Health Sciences University, Molotlegi Street, Ga-Rankuwa, Pretoria 0208, South Africa; 16School of Pharmacy, Sefako Makgatho Health Sciences University, Ga-Rankuwa, Pretoria 0208, South Africa; 17Strathclyde Institute of Pharmacy and Biomedical Sciences, Strathclyde University, Glasgow G4 0RE, UK; 18Centre of Medical and Bio-Allied Health Sciences Research, Ajman University, Ajman 346, United Arab Emirates

**Keywords:** antimicrobial utilization, AWaRe classification, antimicrobial resistance, Pakistan, surgical site infections, surgical wounds

## Abstract

Antimicrobial prophylaxis is effective in reducing the rate of surgical site infections (SSIs) post-operatively. However, there are concerns with the extent of prophylaxis post-operatively, especially in low- and middle-income countries (LMICs). This increases antimicrobial resistance (AMR), which is a key issue in Pakistan. Consequently, we conducted an observational cross-sectional study on 583 patients undergoing surgery at a leading teaching hospital in Pakistan with respect to the choice, time and duration of antimicrobials to prevent SSIs. The identified variables included post-operative prophylactic antimicrobials given to all patients for all surgical procedures. In addition, cephalosporins were frequently used for all surgical procedures, and among these, the use of third-generation cephalosporins was common. The duration of post-operative prophylaxis was 3–4 days, appreciably longer than the suggestions of the guidelines, with most patients prescribed antimicrobials until discharge. The inappropriate choice of antimicrobials combined with prolonged post-operative antibiotic administration need to be addressed. This includes appropriate interventions, such as antimicrobial stewardship programs, which have been successful in other LMICs to improve antibiotic utilization associated with SSIs and to reduce AMR.

## 1. Introduction

Healthcare-associated infections (HAIs) are a common adverse event in hospitals, resulting in increased length-of-stay, morbidity and mortality as well as increased costs [1,2,3,4,5,6,7,8]. In higher-income countries (HIC), device-associated infections are the most common HAI, while among low-and middle-income countries (LMICs), surgical-site infections (SSIs) are the most common HAI [7,9,10,11,12]. Consequently, these patients need to be carefully managed to prevent SSIs, especially in LMICs.

SSIs arise from infective microorganisms, which grow either on the incision site (superficial and deep) or involve an organ or space within 30 days after operation [13,14,15]. They are one of the main problems arising from surgical operations and are the primary source of nosocomial infections in hospital [16,17,18]. SSIs have a considerable burden in terms of their impact on morbidity and mortality as well as increasing costs through increased lengths-of-stay in hospital [4,19,20,21]. The rate of SSIs varies considerably according to the hospital setting, experience of the surgeon, the surgical procedure and the country [7,17,18,22]. Reported prevalence rates vary between 3% to 50% of surgical procedures, with typically appreciably higher prevalence rates among LMICs [11,12,17,23,24,25]. Reported prevalence rates in the United Kingdom and United States of America (USA) for SSIs vary between 16% and 31% [19,26]. Across Africa, combined prevalence rates of 14.8% are seen in surgical patients [10]. In Pakistan, SSI rates can vary from 6.5% up to 33.7% of patients admitted to surgical wards [27,28,29]. Overall, SSIs can account for up to 60% of HAIs in LMICs [1,9,11,30,31]. In view of this, the management of surgical patients requires careful scrutiny to improve future care and reduce costs.

Surgical procedures have been classified into four categories, i.e., Clean, Clean/Contaminated, Contaminated and Dirty, with increasing frequency of bacterial impurity and post-operative infections [15,32,33]. Surgical antimicrobial prophylaxis (SAP) is the administration of antimicrobials before or after surgical procedures to prevent post-operative SSIs [11,17,34].

Appropriate prescription of antibiotics is seen as one of the most effective ways to reduce SSIs [17,18,35,36]. Previous studies and organizations have suggested that antibiotics should ideally be administered 60 min before the first incision, with a second dose potentially administered for longer procedures, i.e., longer than 4 h [36,37,38,39,40,41,42]. Administration prior to this can increase the risk of SSIs by up to five-fold, and delaying the administration until after the first incision can potentially double the risk of an SSI [17,43]. Consequently, this should be avoided. However, alongside the pharmacokinetics and pharmacodynamics of antimicrobials prescribed, their timing, dose and route of administration; possible encountered pathogens; and necessary bactericidal concentrations should also be considered for optimum prophylaxis [35,44,45]. According to World Health Organization (WHO), post-operative SAP should be considered for up to 24 h after the incision is made, where pertinent [46]. More recently, the WHO AWaRe guidance book (2022) has suggested that antibiotic prophylaxis should be administered 120 min or less before starting surgery as a single dose, and not continued after surgery. An additional dose of antibiotics should only be considered where the procedure is prolonged or if there has been major blood loss [33]. Cefazolin, an ‘Access antibiotic’, is recommended as a first choice, and can potentially be combined with other antibiotics, including metronidazole or gentamicin, depending on the procedure [33].

However, extending prophylaxis beyond one day does not reduce SSI rates, but at the same time increases adverse events, antimicrobial resistance (AMR) and costs [17,35,36,39,40,41,47]. Consequently, this also needs to be avoided, especially given the rising AMR rates across countries, particularly among LMICs [48].

A number of studies have shown that the prescription of antibiotics to prevent SSIs is currently not being carried out according to the current guidelines. This includes concerns with the timing of the first dose as well as the selection of the antibiotics and the duration of the prescription [17,18,49]. Typically, prophylaxis extends well beyond the post-operative period, especially among LMICs (Appendix A); however, this is not always the case [17,18,44,50]. The considerable variation seen in practice, especially among LMICs, could be due to several factors. These include differences in published guidelines across countries, lack of awareness of the content of the current guidelines, concerns with the cleanliness of the theaters and wards in the hospital as well as a lack of acceptance of the current guidelines among surgeons [17,18,49,51].

Overall, the irrational prescription of antibiotics, including the excessive use of broad-spectrum antibiotics, is common for surgical procedures. This is especially true among LMICs, including Pakistan [17,18,50,52]. However, there is currently limited available research regarding the prescription of antibiotics for SAP among hospitals in Pakistan that is broken down by the surgical procedures undertaken. This is the first step to developing future quality improvement programs to address current concerns. Consequently, this study was undertaken to observe the utilization patterns of antibiotic prescription during various surgical procedures in a leading tertiary hospital in Pakistan. Key areas of investigation were the choice of antibiotic for given procedures and the timing and duration of the prescription. The findings would be used to provide future guidance for this and other hospitals in Pakistan. Key activities include instigating appropriate antimicrobial stewardship programs (ASPs) to improve future antibiotic prescription if appreciable concerns are found [17,18,29]. This is because a number of ASPs have been successfully introduced among LMICs to improve antibiotic prescription surrounding SSIs, providing direction to all key stakeholders in Pakistan in the future.

## 2. Results

### 2.1. Sociodemographic and Clinical Characteristics of Study Participants

The medical records of 583 surgical patients who were prescribed an antimicrobial as part of SAP were evaluated. The mean age of the patients was 42.2 years. Out of the total study population, 51.3% of the participants were male (Table 1). A minority of the patients had comorbid conditions, and the majority of these suffered from diabetes (4.6% of the surgical patients). The majority of the patients were from the general surgery ward (50%), and most of these were admitted due to high pain scores. Table 1 contains further details regarding the wards of the patients along with the diagnoses (Table 1).

Osteomyelitis (27.4%), followed by cholelithiasis (13.4%) and abscess formation (12.1%), were the most common surgical procedures (Table 1). Regarding the types of surgical procedures which were performed, implant fixation due to the removal of dead bone was the most common procedure, accounting for approximately 27.4% of all procedures, followed by caesarean sections (13.8%) (Table 2). As a result, just under half of the patients had a dirty procedure performed (45.7%), followed by those who received a clean contaminated procedure (30.8%) (Table 2).

### 2.2. Antimicrobial Utilization Patterns for Surgical Prophylaxis

The different antimicrobial classes prescribed as part of SAP for the different types of wound surgery included penicillins (in combination with beta-lactamase inhibitors), cephalosporins (in combination with beta-lactamase inhibitors), aminoglycosides, fluoroquinolones, vancomycin and metronidazole. The most prevalent use of antimicrobials was for dirty procedures (45.7%), followed by clean contaminated wounds (30.7%).

Overall, cephalosporins were the most commonly prescribed type of antibiotic, followed by penicillins. Among the penicillins, co-amoxiclav was extensively prescribed in patients undergoing clean procedures (36.5%), whereas there was appreciable prescription of cefoperazone plus sulbactum (24.4%), of the cephalosporins, in patients undergoing dirty procedures. There was also appreciable prescription of amikacin and vancomycin in patients undergoing dirty procedures. Most of the patients undergoing dirty procedures received a dual combination of antibiotics, e.g., cefoperazone/sulbactum plus amikacin, and in some cases, triple combinations (Table 3). Most of the prescribed antibiotics were in the ‘Watch’ category.

Concerning the timing of antibiotic administration, all the patients in our study were administered their antimicrobials from half an hour to one hour before the surgical procedure.

The duration of antimicrobial prophylaxis was 3.5 ± 1.5 days, typically 3 to 4 days post-operatively. The antibiotics prescribed post-operatively were typically a continuation of those prescribed prior to the first incision. These antibiotics were prescribed post-operatively to prevent SSIs, typically without performing culture sensitivity testing (CST). Overall, antibiotics were considered as prophylactics in clean, clean contaminated and contaminated wounds, and were used for treatment in the case of dirty wounds.

## 3. Discussion

We believe this study is the first to be conducted in Pakistan to assess the pattern of antimicrobial usage for the different types of surgical wounds, i.e., clean, clean/contaminated, contaminated and dirty. Encouragingly, nearly all patients received their antimicrobial prophylaxis half to one hour before the surgical procedure, indicating good compliance with this particular aspect of prophylaxis in the current guidelines and, in addition, in the recent AWaRe guidance [33,35]. This is certainly not universal, as there have been concerns with the timing of prophylaxis pre-operatively as well as extensive prescription post-operatively among a number of LMICs [11,17,18].

However, a number of areas of concern were identified in our study that need to be addressed to improve the future prescription of antimicrobials in surgical patients to reduce AMR and associated costs. Firstly, all the patients in our study subsequently received antimicrobials for their surgical procedures post-operatively. This indicates their overuse in this situation, as there is no need to prescribe antibiotics post-operatively to prevent SSIs, especially for clean procedures [33,35]. The extensive prescription post-operatively until the day of discharge that was seen in our study in Pakistan may be due to a number of factors. These include a lack of awareness and non-compliance with current international guidelines among physicians, i.e., perceiving that such guidelines are not directly attributable to their hospital and setting, concerns with the cleanliness of their wards, a general lack of infection control practices due to resource issues as well as a lack of trust in the validity of antimicrobial susceptibility data in the hospital [17,53,54,55]. Our findings regarding the extensive prescription of antimicrobials post-operatively for SAP are similar to those of a study by Khan et al. (2020), as well as one by Saleem et al. (2023), in Pakistan alongside other LMICs (Appendix A), and this issue urgently needs to be addressed going forward [55,56]. Our findings are different from those of a recent study in Ethiopia, where 97.4% of patients who were administered antibiotics for SAP were prescribed a single dose. This typically occurred one hour before surgical procedures (in 88% of patients undergoing SAP) [50]. Successful ASPs have been undertaken in a range of LMICs to improve SAP, thereby providing exemplars to key stakeholder groups in Pakistan for the future. These are summarized in Appendix A.

Another concern was the frequent use of third generation cephalosporins in this study, which are typically ‘Watch’ antibiotics under the WHO AWaRe classification [57,58]. Unnecessary prescription of ‘Watch’ and ‘Reserve’ antibiotics should be reduced as part of an effort to reduce resistance rates within a country [59,60,61]. This is particularly important in Pakistan given the high rates of AMR in the country, generally high rates of prescription of ‘Watch’ and ‘Reserve’ antibiotics within hospitals and the urgent need to reduce AMR as part of the agreed National Action Plan [52,56,62,63,64,65]. The National Institute of Health (NIH) of Pakistan has recognized the impact of HAIs-initiated activities attempting implement infection control training programs and provide the necessary infrastructure to support prevention programs as part of the agreed National Action Plan (NAP) to reduce AMR in Pakistan [62,65]. This is encouraging, since a number of studies have now shown that a range of interventions as part of ASPs can appreciably improve the use of antimicrobials to prevent SSIs in LMICs (Appendix A). We are aware that ASPs have begun to be introduced across LMICs, including Pakistan; however, challenges remain [53,66]. The plethora of ASPs that have now been introduced across LMICs to reduce the inappropriate use of antibiotics as part of SAP, including among hospitals in Pakistan (Appendix A), can serve as exemplars to key stakeholder groups across Pakistan. We will continue to monitor this as a key activity to reduce AMR in hospitals in Pakistan going forward. In addition, the guidance for SAP advocated in the recent AWaRe Guidance book should be promoted. As such, continued extensive prescription of ‘Watch’ antibiotics for SAP in Pakistan should be reduced, alongside prolonged prescriptions [33]. In addition, regularly recording patient outcomes in their notes is essential.

We are aware that there are a number of limitations within our study. Firstly, we only involved one center in this study. However, this center was carefully chosen as an exemplar for a tertiary hospital in Pakistan. Secondly, we only selected patients in the orthopedic, general surgery and gynecology wards, as we were mainly interested in current antimicrobial prescribing patterns to prevent SSIs. Culture reports were also not checked to identify resistance patterns in the hospital. Consequently, we were unable to assess the appropriateness of the antibiotics prescribed, including third generation cephalosporins, based on current sensitivity patterns. We also did not check the actual prevalence of SSIs due to a lack of patient follow-up details in patients’ notes. Lastly, these observational studies, by their very nature, are likely to involve missing key information sets. In addition, it is difficult to identify true risk factors. Nevertheless, we believe that our findings are robust and will provide a direction for all key stakeholder groups in Pakistan in the future.

## 4. Materials and Methods

### 4.1. Study Setting and Design

This observational, cross-sectional study was conducted in the orthopedic, general surgery and gynecological wards of Ghurki Trust Teaching Hospital (GTTH), Lahore, Pakistan, to explore the utilization pattern of antimicrobials in surgical patients with clean, clean/contaminated, contaminated and dirty wounds, as defined by ACS-NSQIP (Table 4) [67]. This hospital was chosen for this initial study as it is a leading teaching hospital in Lahore. GTTH is a purpose-built tertiary care hospital located in Lahore, Pakistan. The hospital was established in 2011 by the Ghurki Trust, which aims to provide high-quality healthcare services. The hospital has a capacity of 600 beds and offers a wide range of medical and pharmacy services. It is equipped with state-of-the-art medical equipment and technology to provide the latest and most advanced treatments to its patients. Consequently, if there are issues in this hospital, they are likely to be replicated in other hospitals throughout Pakistan.

All the patients admitted to the gynecologic, general surgical and orthopedic wards who underwent clean, clean/contaminated, contaminated and dirty procedures were included. These wards were carefully chosen to provide the maximum number of patients for the study. Seriously ill patients subsequently requiring admission into the Intensive Care Unit (ICU) and patients admitted to other wards in the hospital were excluded from the study.

### 4.2. Data Collection Process, Questionnaire and Statistical Analysis

Data were collected from 583 patients who underwent surgery at the admitted hospital. Only antimicrobials used during hospitalization were recorded. Antimicrobials were typically not switched to a peroral form or to a narrow spectrum antibiotic during this period.

The data collection form that was utilized in this study was based on previous publications, combined with the considerable experience of the co-authors [11,13,17,20,23,28,30]. We have used this approach before for similar research projects across different countries [11,64,68].

The form was divided into four key sections. These included:Patient demographic and medical data, e.g., age, gender, co-morbidities, chief complaint, diagnosis, length of hospital stay and duration of antibiotic prescription;Surgical data, i.e., type of surgery;Antimicrobial utilization data, e.g., name, antimicrobial class (ATC code), AWaRe classification, frequency and duration of administration [57,58,69]. Under the AWaRe classification, antibiotics in the ‘Access’ group should be used against commonly encountered infections, as they have a lower resistance rate, while those in the ‘Watch’ group should only be used in critical conditions, as they have a greater chance of resistance development. Antibiotics in the ‘Reserve’ group should only be prescribed in multi-drug resistance cases [57,58];Wound classification (clean, clean/contaminated, contaminated or dirty).

All the collected data were analyzed using the Statistical Package for Social Sciences (SPSS) and the latest versions of Microsoft Excel. Descriptive statistics were used to summarize the patient demographics and clinical data.

### 4.3. Antimicrobial Stewardship Programs to Reduce Inappropriate Antibiotic Prescription for SAP among LMICs

This will be a narrative review of the literature rather than a systematic review to provide guidance to key stakeholder groups in Pakistan. This is because there have been a number of published reviews to date surrounding the extensive prescription of antibiotics post-operatively as part of SAP, as well as other potential interventions to reduce SSIs [17,18,44,70,71,72,73,74,75,76].

Alongside this, our objective was to document ASPs that have been instigated to improve future SAP as part of the discussion, and to provide exemplars that could subsequently be used to guide key stakeholders in Pakistan regarding potential ways forward to improve appropriate SAP techniques when there are concerns. This builds on the current examples used in Pakistan. In light of this, we performed a narrative review of pertinent publications rather than undertaking a systematic review. We have successfully used this approach before in LMICs to stimulate debate surrounding different key disease areas and topics [77,78]. Consequently, we believed that this methodology was appropriate for this study.

### 4.4. Ethical Approval

The study was approved by the Human Ethics Committee of College of Pharmacy, Bahauddin Zakariya University, Multan, Pakistan (BZU-DEPP-22-830-1006). The data were collected after obtaining approval from the management of the Ghurki Trust Teaching Hospital.

## 5. Conclusions

In conclusion, the patterns of antibiotic utilization in this hospital were not in full accordance with the current standard guidelines, including the AWaRe guidance, to prevent SSIs. The overuse of ‘Watch’ antibiotics, as well as the prolonged duration of antibiotic prescription post-operatively, are concerns that need to be addressed going forward to reduce AMR in hospitals in Pakistan. In light of this, there is an urgent need to promote pertinent ASPs throughout Pakistan to improve future antimicrobial prescription for SAP. These can be based on ASPs that have been successfully implemented in other LMICs. We will continue to monitor the situation.

## Figures and Tables

**Table 1 antibiotics-12-00678-t001:** Sociodemographic characteristics and medical conditions.

Variable	Frequency (N = 583)
Demographics	
Age	42.2 ± 18.4
Sex	
Male	299 (51.3%)
Female	284 (48.7%)
Comorbidities	
Diabetes mellitus	27 (4.6%)
Hypertension	16 (2.7%)
Hepatitis	14 (2.4%)
Heart disease	5 (0.9%)
Clinical Presentation	
Ward	
General surgery	268 (50%)
Orthopedic	214 (36.7%)
Gynecological	101 (17.3%)
Top Three Complaints on Hospital Admission	
Pain	70 (12%)
Swelling	45 (7.7%)
Pus discharge	12 (2.1%)
Diagnosis	
Bone Infections	
Osteomyelitis	160 (27.4%)
Internal Fracture	54 (9.2%)
Intra-abdominal Infections	
Cholelithiasis	78 (13.4%)
Hernia	50 (8.6%)
Appendicitis	33 (5.7%)
Full term delivery	80 (13.8%)
Fibroids	21 (3.7%)
Skin and Soft Tissue Infections	
Abscess	71 (12.1%)
Diabetic foot	36 (6.1%)
Hospital Stay	
Total duration in hospital (days), including pre-operatively	4.9 ± 3.8

**Table 2 antibiotics-12-00678-t002:** Surgical procedure performed.

Surgical Procedures	Frequency (%)
Implant Fixation Due to Removal of Dead Bone (Dirty Wound)	160 (27.4)
Caesarean section (Clean Contaminated Wound)	80 (13.8)
Lap Cholecystectomy (Clean Contaminated Wound)	78 (13.4)
Abscess Drainage (Dirty Wound)	71 (12.3)
Implant Fixation Due to Fracture (Clean Wound)	54 (9.2)
Hernia Repair (Clean Wound)	50 (8.6)
Amputation Due to Diabetic Foot Necrosis (Dirty Wound)	36 (6.1)
Appendectomy (Contaminated Wound)	33 (5.6)
Hysterectomy (Clean Contaminated Wound)	21 (3.6)

**Table 3 antibiotics-12-00678-t003:** Utilization pattern of antibiotics among surgical patients.

Antibiotics	WHO AwaRe Classification	Wound Types				Total N (%)
		Clean N (%)	Clean/Contaminated *n* (%)	Contaminated *n* (%)	Dirty *n* (%)	
Penicillins						
Piperacillin + Tazobactam (J01CR05)	Watch	2 (1)	25 (7.1)	2 (4)	111 (20.1)	140 (12.1)
Co-amoxiclav (J01CR02)	Access	38 (19)	18 (5.1)	12 (24.4)	5 (0.9)	73 (6.3)
Cephalosporins						
Cefoperazone + Sulbactam (J01DD62)	Watch	20 (10)	78(22.2)	12 (24.4)	120 (21.8)	230 (20)
Ceftriaxone (J01DD04)	Watch	10 (5)	15 (4.2)	6 (12.2)	61 (11)	92 (8)
Cefazolin (J01DB04)	Access	40 (20)	26 (7.4)	-	-	66 (5.7)
Cefuroxime (J01DC02)	Watch	10 (5)	30 (8.5)	-	10 (1.8)	50 (4.3)
Cephradine (J01DB09)	Access	22 (11)	8 (2.2)	-	-	30 (2.6)
Cefixime (J01DD08)	Watch	16 (8)	13 (3.7)	-	-	29 (2.5)
Fluoroquinolones						
Ciprofloxacin (J01MA02)	Watch	16 (9)	30 (8.5)	8 (16.2)	3 (0.5)	59 (5.1)
Moxifloxacin (J01MA14)	Watch	4 (2)	20 (5.7)	3 (6.1)	-	27 (2.3)
Aminoglycosides						
Amikacin (J01GB06)	Access	8 (4)	26 (7.4)	-	120 (21.8)	154 (13.4)
Others						
Metronidazole (J01XD01)	Access	12 (6)	61 (17.4)	4 (8.1)	95 (17.2)	172 (14.9)
Vancomycin (J01XA01)	Watch	-	-	2 (4)	25 (4.5)	27 (2.3)
Total antimicrobials		200 (17.4)	350 (30.4)	49 (4.2)	550 (47.8)	1149 (100)

NB: ‘A’ = access antibiotic, ‘W’ = watch antibiotic according to the AWaRe classification (see Section 4.2 for details).

**Table 4 antibiotics-12-00678-t004:** Surgical wound classifications by ACS-NSQIP *.

Wound Type	Definition
Clean	Uninfected operative wounds without inflammation or involvement of the respiratory, alimentary, genital or uninfected urinary tracts
Clean/Contaminated	Operative wounds in which the respiratory, alimentary, genital or urinary tract is entered under controlled conditions and without unusual contamination
Contaminated	Open, fresh, accidental wounds; operations with major breaks in sterile technique or gross spillage from the gastrointestinal tract; and incisions in which acute, non-purulent inflammation is encountered
Dirty	Old traumatic wounds with retained devitalized tissue and involvement of existing clinical infection or perforated viscera

NB: * American College of Surgeons National Surgical Quality Improvement Program.

## Data Availability

Additional data are available upon reasonable request to the corresponding authors.

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
