# Peer review of "Antibiotic Utilization Patterns for Different Wound Types among Surgical Patients: Findings and Implications"

_antibiotics, 2023, doi:10.3390/antibiotics12040678_

Round 1

Reviewer 1 Report

Thank you for the opportunity to review this article. This retrospective, observational study evaluated the surgical antimicrobial prophylaxis prescribing patterns in a large, academic hospital in Pakistan. This study is timely as antimicrobial resistance poses an urgent threat for the medical community and overuse of antimicrobial prophylaxis in low- middle-income countries (LMIC). This study evaluates prescribing habits for SAP as well as reviews literature for ASP interventions in SAP for LMICs.

Overall, I commend the investigators for evaluating an important antimicrobial stewardship topic. The retrospective nature of the study provides several significant limitations, which are disclosed in the body of the discussion. I have a few questions and comments outlined below that I believe should be addressed to improve the validity and scope of this manuscript.

I have a few general questions and comments summarized below:

Grammar:

-        This manuscript would benefit from further editing for grammatical purposes, there were several run on sentences, misplaced words, and punctuation errors.

Abstract:

-        Line 47-49: run on sentence, consider revising

-        Line 51: I’m not sure “concerns” is the correct word here. Maybe variables?

-        Line 54 – 55: Adding “concerns” may be conceived as adding bias to the manuscript. Would recommend simply stating the findings surrounding durations of antimicrobial prophylaxis compared to guideline recommended durations.

Introduction:

-          Line 65-66: Run on sentence, consider revising

-          Line 69-70: should managed be changed to “prevented?”

-          Line 74-76: would say increasing costs and increased length of stay.

-          Line 79-82: run on, consider revising

-          Line 95: consider removing “including 120 minutes before first incision”

-          Line 114-116: run on, consider revising

-          Certain aspects of the introduction are worded very strongly and may benefit from softening tone to avoid a defensive posture for the readers of non-ASP backgrounds.

Results:

-          Line 139: states most patients were suffering from diabetes, however, looking at Table 1, it appears only 4.6% had diabetes.

-          Would consider reformatting Table 1 to break up into sections such as demographics, treatment location, presenting symptoms, and diagnosis on admission.

-          For diagnosis: would consider grouping based on type (intra-abdominal with sub-lines to states specific diagnosis)

-          For “days after surgery” is that days of antibiotics or days of admission?

-          Line 148: similar to diabetes comment, would modify the word “most” as osteo only comprised 27% of the population.

-          Table 2, could be integrated into Table 1

-          Likewise, Table 3 could be integrated into Table 1. Similar to “diagnosis” section, would group surgical procedures by general type and then sub-categorize.

-          Line 165: Penicillin should be the plural penicillins not the possessive penicillin’s

-          Line 181-183: is this stating that all patients received antimicrobials prior to surgery timed appropriately?

-          Line 187-189: Does this mean that antimicrobials used before/after any dirty procedure was considered treatment rather than prophylaxis?

-          Section 2.3: The type and scope of this manuscript becomes a little unclear here. It begins as an observational review of antimicrobial prophylaxis prescribing at a hospital, then transitions here to a review article summarizing ASP interventions in LMICs.

Discussion:

-          I think this manuscript would benefit from adding data on the rates of appropriate antimicrobial prophylaxis selection, guideline adherence, durations, rates of antimicrobial resistance, and infection rates.

-          Line 211-213: what percentage of patients had antibiotics stopped prior to discharged?

-          Line 224-225: Was appropriateness of third generation cephalosporin use assessed based on patient-specific risk factors or procedure SAP recommendations?

-          Line 227-229: Was culture or previous infectious history included in evaluating appropriateness of prophylaxis. Given the high rates of osteomyelitis or intra-abdominal infections, current or previous infections may have impacted the recommended antimicrobial selection.

-          Line 245-248: I think this manuscript would greatly benefit from SSI or subsequent AMR data.

-          I believe the limitations need to be expanded based on the observational nature of the study, limited data to classify appropriateness of antimicrobial prophylaxis, and lack of clinical outcomes data.

Methods:

-          Would consider moving above results and discussions to set framework for manuscript.

Author Response

Comments and Suggestions for Authors

1) Thank you for the opportunity to review this article. This retrospective, observational study evaluated the surgical antimicrobial prophylaxis prescribing patterns in a large, academic hospital in Pakistan. This study is timely as antimicrobial resistance poses an urgent threat for the medical community and overuse of antimicrobial prophylaxis in low- middle-income countries (LMIC). This study evaluates prescribing habits for SAP as well as reviews literature for ASP interventions in SAP for LMICs.

Overall, I commend the investigators for evaluating an important antimicrobial stewardship topic. The retrospective nature of the study provides several significant limitations, which are disclosed in the body of the discussion.

Author comments: Thank you for these kind words – greatly appreciated.

2) I have a few questions and comments outlined below that I believe should be addressed to improve the validity and scope of this manuscript.

Author comments. Thank you for your comments. Hopefully, we have adequately addressed these.

I have a few general questions and comments summarized below:

3) Grammar:

  1. a) This manuscript would benefit from further editing for grammatical purposes, there were several run on sentences, misplaced words, and punctuation errors.

Abstract:

  1. b) Line 47-49: run on sentence, consider revising

Author comments: Thank you - now revised

  1. c) Line 51: I’m not sure “concerns” is the correct word here. Maybe variables?

Author comments: Thank you - now revised

  1. d) Line 54 – 55: Adding “concerns” may be conceived as adding bias to the manuscript. Would recommend simply stating the findings surrounding durations of antimicrobial prophylaxis compared to guideline recommended durations.

Author comments: Thank you - now revised

Introduction:

-          Line 65-66: Run on sentence, consider revising

Author comments: Thank you - now revised

-          Line 69-70: should managed be changed to “prevented?”

Author comments: Thank you - now revised

-          Line 74-76: would say increasing costs and increased length of stay.

Author comments: Thank you - now revised

-          Line 79-82: run on, consider revising

Author comments: Thank you - now revised

-          Line 95: consider removing “including 120 minutes before first incision”

Author comments: Thank you - now revised

-          Line 114-116: run on, consider revising

Author comments. Thank you – we would like to keep this as however …, etc., forms part of this complete sentence. We hope this is OK.

-          Certain aspects of the introduction are worded very strongly and may benefit from softening tone to avoid a defensive posture for the readers of non-ASP backgrounds.

Author comments: Thank you - now hopefully revised – although want the authorities across LMICs to seriously look at their SAP as considerable over use currently across countries alongside rising concerns with AMR rates, etc. We subsequently discuss ASPs that should be introduced to improve antimicrobial prescribing and subsequently reduce AMR and costs. We hope this is now acceptable.

Results:

-          Line 139: states most patients were suffering from diabetes, however, looking at Table 1, it appears only 4.6% had diabetes.

Author comments: Thank you – now revised

-          Would consider reformatting Table 1 to break up into sections such as demographics, treatment location, presenting symptoms, and diagnosis on admission.

Author comments: Thank you - now revised

-          For diagnosis: would consider grouping based on type (intra-abdominal with sub-lines to states specific diagnosis)

Author comments: Thank you – now undertaken. We hope this is acceptable.

-          For “days after surgery” is that days of antibiotics or days of admission?

Author comments: Thank you for your comment. We have now removed this as causes confusion when later on we discuss the length of antibiotic prescribing post-operatively. We hope this is now acceptable

-          Line 148: similar to diabetes comment, would modify the word “most” as osteo only comprised 27% of the population

Author comments: Thank you – now changed

-          Table 2, could be integrated into Table 1

Author comments: Thank you – we have now extended Table 2 by merging this with key datasets from Table 3. We hope this is now OK.

-          Likewise, Table 3 could be integrated into Table 1. Similar to “diagnosis” section, would group surgical procedures by general type and then sub-categorize.

Author comments: Thank you for your comment – we have now merged the Tables.

-          Line 165: Penicillin should be the plural penicillins not the possessive penicillin’s

Author comment: Thank you – now revised

-          Line 181-183: is this stating that all patients received antimicrobials prior to surgery timed appropriately?

Author comment: Thank you for your comment. This was the observed pattern in our study.

-          Line 187-189: Does this mean that antimicrobials used before/after any dirty procedure was considered treatment rather than prophylaxis?

Author comment: Thank you for your comment. In case of dirty wounds, antibiotics were prescribed before the surgery were considered as treatment rather than prophylaxis. We hope this is now OK.

-          Section 2.3: The type and scope of this manuscript becomes a little unclear here. It begins as an observational review of antimicrobial prophylaxis prescribing at a hospital, then transitions here to a review article summarizing ASP interventions in LMICs.

Author comments: Thank you – this is correct as we wanted to include guidance to key stakeholders in Pakistan. However – we have now moved this Table to the Discussion to avoid any confusion. We hope this is now acceptable.

Discussion:

-          I think this manuscript would benefit from adding data on the rates of appropriate antimicrobial prophylaxis selection, guideline adherence, durations, rates of antimicrobial resistance, and infection rates.

Author comments: Thank you for this. We have added this data where we can since, as mentioned, some of this data is currently unavailable. We hope this is OK with you

-          Line 211-213: what percentage of patients had antibiotics stopped prior to discharged?

-          Line 224-225: Was appropriateness of third generation cephalosporin use assessed based on patient-specific risk factors or procedure SAP recommendations?

-          Line 227-229: Was culture or previous infectious history included in evaluating appropriateness of prophylaxis. Given the high rates of osteomyelitis or intra-abdominal infections, current or previous infections may have impacted the recommended antimicrobial selection.

-          Line 245-248: I think this manuscript would greatly benefit from SSI or subsequent AMR data

Author comments: Thank you for this. Unfortunately, we do not have this data especially AMR data as this is currently typically scarce in Pakistan. In addition – currently limited sensitivity testing in hospitals due to the costs involved (we have seen this in other studies some of the co-authors have undertaken in Pakistan). We are now trying to rectify this for future studies – building on the issues and concerns that we found in this study. We hope this is acceptable. 

-          I believe the limitations need to be expanded based on the observational nature of the study, limited data to classify appropriateness of antimicrobial prophylaxis, and lack of clinical outcomes data.

Author comments: Thank you – now done. We hope this is now acceptable. 

Methods:

-          Would consider moving above results and discussions to set framework for manuscript.

Author comments: Thank you for this – this is the current template supplied by the Journal.

Reviewer 2 Report

In this manuscript the authors have undertaken a study to record the antibiotic utilization patterns in patients undergoing orthopedic, gynecologic, general surgery in a teaching hospital at Lahore. Their findings indicate that there is an excessive use of "Watch" antibiotics that should be corrected to reduce antimicrobial resistance problem in the future.

Line 73: One of "the" main

Line 130: providing future guidance. "future" is redundant

Line 139: % of diabetes mellitus patients in the table vs what is described in the text does not corelate very well. Please clarify.

Line 150, 151: Does not reflect in the Table 

Line 203 Vs Line 210: How does this match ? Please clarify.

Line 221-223: What is the relevance of this information. How does it tie to your findings. Please explain.

Line 245: "within" our study.

Line 267" "if" there are issues

Author Response

Comments and Suggestions for Authors

1) In this manuscript the authors have undertaken a study to record the antibiotic utilization patterns in patients undergoing orthopedic, gynecologic, general surgery in a teaching hospital at Lahore. Their findings indicate that there is an excessive use of "Watch" antibiotics that should be corrected to reduce antimicrobial resistance problem in the future.

Author comments: Thank you for this summary – appreciated.

2) Line 73: One of "the" main

Author comments: Thank you now changed

3) Line 130: providing future guidance. "future" is redundant

Author comments: Thank you now changed

4) Line 139: % of diabetes mellitus patients in the table vs what is described in the text does not corelate very well. Please clarify.

Author comments Thank you now changed

5) Line 150, 151: Does not reflect in the Table 

Author comments: Thank you – now changed.

Line 203 Vs Line 210: How does this match? Please clarify.

Author comments: Thank you – now clarified. We hope this is now acceptable.

Line 221-223: What is the relevance of this information. How does it tie to your findings. Please explain.

Author comments: Thank you – now made more explicit. As seen, we have also moved the old Table with ASP examples to Supplementary Material in line with the suggestions from one reviewer.

Line 245: "within" our study.

Author comments: Thank you – now changed.

Line 267" "if" there are issues

Author comments: Thank you – now changed.

Round 2

Reviewer 1 Report

Thank you again for your efforts in completing this research endeavor. I appreciate your responses and edits to my comments. I believe the modifications made were appropriate. 

Reviewer 2 Report

The authors have made necessary changes and the manuscript is now satisfactory.

Line 142: no need of "and" after "A minority of"